# Results from a maternal, newborn and child health program targeting pregnant, married adolescent girls in northern Nigeria

Erica Felker-Kantor[1], Mary Phillips[1*], Meghan Cutherell[1], Roselyn Odeh[2], Stacy Lois[2], Alhaji Bulama[2], Abednego Musau[1]

**1** Population Services International, Washington, DC, United States of America, **2** Society for Family Health, Abuja, Nigeria

\* mphillips@psi.org

## Abstract

Adolescent pregnancy is a significant public health concern in Nigeria where adolescent girls (15–19 years) face high rates of maternal morbidity and mortality compared to older women. Interventions that address the unique needs of pregnant adolescents in the Nigerian context are critical to improve pregnancy-related health outcomes. This paper presents results from a longitudinal quasi-experimental study evaluating the effectiveness of a maternal, newborn and child health (MNCH) intervention on pregnancy and postpartum outcomes among married adolescent girls in northern Nigeria. The study included an intervention and comparison group and was conducted in Kaduna and Jigawa states. Data were collected at baseline, 3-months post intervention, and at 8-weeks postpartum. Results from the cross-sectional postpartum survey are presented in this paper. The postpartum survey sample included 846 married adolescent girls aged 15–19, 392 (46%) in the comparison group and 454 (54%) in the intervention group. Compared to adolescent girls in the comparison group, those exposed to the intervention were more likely to attend any antenatal care visit (IRR: 1.72; 95%CI: [1.56, 1.90]), attend antenatal care at an earlier gestational age (β: -0.54; 95%CI: [-0.89, -0.21]), attend more antenatal care visits (β: 1.39; 95%CI: [1.01, 1.77]), use nutritional supplements during pregnancy (IRR: 1.49; 95%CI: [1.35, 1.64]), give birth at a health facility (IRR: 2.44; 95%CI: [2.48, 4.05]), use postpartum family planning (IRR: 3.17; 95%CI: [2.48, 4.05]), and receive postnatal care (IRR: 1.48; 95%CI: [1.25, 1.76]). The adolescent-friendly MNCH intervention demonstrated positive and statistically significant effects on pregnancy and postpartum outcomes among married adolescent girls in northern Nigeria. Involving adolescents in program design played an important role in creating a program that addressed the specific needs of pregnant adolescent girls. Scaling up such interventions to improve health outcomes for pregnant adolescent girls can impact their immediate health but also their future health trajectories and that of their children.

**Data availability statement:** The data supporting the findings in this study are publicly available in Figshare under a CC BY 4.0 license under the title "Postpartum Survey Tool and Data set" and DOI: 10.6084/m9.figshare.29574227.

**Funding:** The study was funded by the Gates Foundation (INV-004274) and Children Investment Fund Foundation (R-1911-04245). The funders had no role in study design, data collection and analysis, decision to publish, or preparation of the manuscript.

**Competing interests:** The authors have declared that no competing interests exist.

## Background

Complications from pregnancy and childbirth are among the leading causes of death for adolescent girls aged 15–19 globally [1]. While adolescent pregnancy rates have decreased, an estimated 21 million adolescent girls give birth each year in low- and middle-income countries (LMICs) [2]. Adolescent mothers often face heightened risk for eclampsia and systemic infections, and babies born to adolescent mothers have an increased risk of premature birth and low birth weight [3,4]. This increased risk may be explained by physiological differences during adolescence and physical immaturity compared to older women as well as demographic and socio-economic characteristics [5]. Ensuring that pregnant adolescent girls have access to quality maternal and newborn health services is critical for mitigating the risks that come with adolescent pregnancy.

There is strong evidence that antenatal care (ANC) reduces pregnancy risks and improves outcomes for all pregnant women [1]. Initiating care early in pregnancy is also associated with improved maternal health and fewer adverse birth outcomes or interventions in late pregnancy [6]. Women who attend ANC receive health education on pregnancy risk factors, nutrition during pregnancy, and are more likely to deliver in a health facility with a skilled birth attendant [7]. Although evidence on the linkage between receiving ANC and attending postnatal care (PNC) and postpartum family planning (PPFP) remains mixed, ANC visits and presence of a skilled provider during birth provide an opportunity to educate women about the health benefits of PNC, birth spacing, and contraceptive use after child birth [8].

In 2016, the World Health Organization (WHO) changed the number of recommended ANC visits from 4 to 8, following evidence which showed that a higher frequency of antenatal contacts with a skilled provider reduced the likelihood of stillbirth [9,10]. Despite the benefits of ANC being well documented, timely uptake of ANC among adolescents in LMICs remains low due to many factors including limited autonomy with respect to decision-making, lack of awareness, financial resources, transportation restrictions, social norms, lack of age-appropriate health services and for married adolescent girls, power imbalances and mobility or permission restrictions [11]. Compared to their older counterparts, adolescents also have significantly lower coverage of PNC due to similar factors [12].

A recent systematic review by Sabet et al. reported on evidence-based interventions to improve health outcomes including uptake of ANC, PNC and use of nutritional supplements for pregnant women and adolescent girls in LMICs [13]. Results found that most interventions were not specific to adolescents, and there was limited quantitative evidence specifically related to improving pregnancy care, delivery, abortion, or PNC for adolescents, and no interventions focused on PPFP uptake. Two of the more effective but resource intensive interventions focused on increasing health facility births through conditional cash transfer programs [14,15]. According to the authors, however, one of the largest gaps in the body of literature was the lack of programming focused on pregnancy-related health outcomes informed by adolescent-friendly approaches. A critical aspect of adolescent healthcare is

recognizing adolescents as partners for designing creative and effective health programming to address their unique needs within their specific cultural contexts.

## Nigeria

Nigeria has the largest population in Africa with an estimated 205 million inhabitants and half the population under the age of 24 [16]. The average age of marriage for Nigerian women is 19 and nearly one in five adolescent girls give birth to their first child between ages 15–19 [17]. In northwest Nigeria, 29% of adolescent girls have started childbearing by the age of 19 compared to just 6% in the southwest [16,17]. The proportion of pregnant women receiving ANC from a skilled provider increased nationally from 58% to 67% between 2008 and 2018 before decreasing to 63% in 2023. Adolescent girls are less likely to receive ANC from a skilled provider compared to other age groups (age 15–19: 57%; age 20–34: 69%; age 35–49: 65%) [18]. Regional differences in ANC also exist, with the lowest rates of ANC occurring in the northwest (54%) and the highest in the southeast (89%) [18]. Multiple factors may contribute to why adolescents are less likely to access and receive ANC including financial constraints, cultural and religious beliefs (especially in northern Nigeria), minimal autonomy, lack of awareness and low education, and distance to a health facility, among others. According to the WHO, the country contributes the largest number of obstetric and postpartum deaths of any country. The national maternal morality ratio (MMR) is 512 maternal deaths per 100,000 live births and the lifetime risk of maternal death indicates that 1 in 34 women will have a death related to maternal causes [11,19]. The highest rate of maternal death occurs in the northwest where 60% of women deliver at home. Adolescent girls aged 15–19 have the second highest MMR in the country just behind young women aged 20–24, with an MMR of 30% [18].

In 2016, with funding from the Gates Foundation and the Children's Investment Fund Foundation (CIFF), Population Services International (PSI) launched Adolescents 360 (A360), to increase demand for, and voluntary uptake of modern contraception among girls aged 15–19 in Ethiopia, Tanzania, and Nigeria [20]. In Nigeria, Society for Family Health leads the implementation of A360's intervention called *Matasa Matan Arewa* (MMA) in four northern states, where there are high rates of adolescent pregnancy: Kaduna, Nasarawa, Jigawa, and Kano. MMA was designed using a human-centered design (HCD) approach, incorporating insights and feedback from married girls living in the region. Implemented within local primary health structures, MMA mobilizes married adolescent girls to receive adolescent-friendly contraceptive services at nearby health centers. The intervention also offers girls the chance to attend a series of four, two-hour mentored group sessions called Life, Family, Health (LFH). LFH sessions cover topics such as sexual and reproductive health (SRH), family nutrition, birth spacing, interpersonal communication, goal setting and financial management [20]. As part of MMA, men are also engaged by male interpersonal communication agents (IPCAs) who facilitate discussions about contraceptive use for married adolescent girls at spots where men congregate and distribute referral cards that men can provide to their wives to access services. Based on a recent evaluation of MMA's effectiveness, married adolescent girls exposed to the intervention are more likely to use modern contraception, have lower unmet need and have more positive attitudes toward modern contraception use compared to those not exposed MMA [21]. In 2021, A360 received additional funding to augment the existing intervention with MNCH component. This MNCH component was co-designed with adolescents using an HCD approach to address barriers to healthy pregnancies including a lack of planning for pregnancy, cultural norms that discourage early ANC attendance, general uncertainty about the value of ANC, low uptake of PPFP, and agency to seek care (please see Supplemental Information, S2 File. *Human Centered Design Process* for further explanation of the HCD process). The MNCH component includes an educational session on pre-conception care, nutrition, ANC, delivery, PNC, and PPFP for girls at risk for pregnancy and pregnant girls. Girls attending this session learn how to prepare for a healthy pregnancy, the benefits of attending ANC and having a facility-based birth, and the importance of PNC and PPFP for birth spacing. The sessions are facilitated by trusted community female mentors trained on the benefits of ANC and PNC, facility-based delivery, and PPFP. Government ANC providers also attend the sessions to provide clinical information

and to build trust with the girls. At the end of the sessions, pregnant girls are provided with pregnancy journey maps based on their gestational age at the time of enrollment. The girls are instructed to take the journey map home and complete the information about their pregnancy and birth plan with their husbands. Husbands are also targeted with messaging from male IPCAs on the importance of ANC during pregnancy and to advocate for their involvement in supporting their wife's health-seeking behaviors during pregnancy. At the facility level, ANC providers receive informational sessions about MMA's MNCH component as well as reminders on how to provide adolescent-friendly services, high-quality ANC, and record-management for effective client tracking. Finally, pregnant girls receive follow-up visits from the trained female mentors to help with ANC retention and encourage facility-based birth.

To evaluate the effectiveness of the MNCH component, we conducted a longitudinal quasi-experimental study in Kaduna and Jigawa states in northern Nigeria. The study included a baseline survey (pre-intervention), 3-month follow-up survey (post-intervention), and an 8-week postpartum survey (pregnant girls only). This paper presents results from the postpartum survey. Results from the baseline and 3-month follow-up surveys are presented elsewhere.

## Materials and methods

### Setting

Kaduna state has a population of approximately nine million [22]. Kaduna has a high fertility rate with an average of 5 children per woman of reproductive age (WRA) [17]. The median age for first marriage among women is 16 and for first birth 18. Slightly more than 30% of girls aged 15–19 have begun childbearing and 13% of pregnancies end in abortion [17]. Maternal mortality is high in Kaduna with an MMR of 452.6 deaths per 100,000 live births [23]. Jigawa state has an estimated population of 7.5 million with many internally displaced populations due to Boko Haram conflicts in a neighboring region. The fertility rate is 7 children per WRA [17]. The median ages for first marriage and first birth are 15.8 and 18, and 45% of marriages are polygamous. Slightly more than 29% of girls aged 15–19 have begun childbearing. Maternal mortality is particularly high with an estimated MMR of 1,012 deaths per 100,000 live births [24].

### Study design

The evaluation of the MNCH component employed a quasi-experimental design with an intervention and comparison group. In Jigawa, five local government areas (LGAs) where MMA is implemented were selected as study locations, three as intervention sites and two as comparison sites. In Kaduna, two LGAs were selected as intervention sites and two as comparison sites. Selection of LGAs was conducted using purposive sampling, ensuring that intervention and comparison sites were separated by substantial geographic distance to reduce spillover effects. The intervention and comparison groups were similar in terms of demographics, socio-economic status and health indicators. The intervention group received the integrated MMA intervention, inclusive of both its original SRH and new MNCH components. The comparison group received the original SRH-focused MMA intervention and MNCH standard of care per MOH guidelines. The primary target population was married adolescent girls aged 15–19 who were at risk for pregnancy, were pregnant, or became pregnant by the 3-month follow-up. To be eligible for the study girls were required to be aged 15–19, married, residing in the study area not using a contraceptive method (non-pregnant girls) or less than 25 weeks pregnant (pregnant girls), not having attended ANC for current pregnancy (pregnant girls), and to provide written informed consent. The study opted to compare the MNCH component outcomes against outcomes resulting from implementation of the traditional SRH-focused MMA intervention. Results were intended to measure incremental effectiveness of layering ANC-focused education and support on top of the core SRH content for pregnant or at risk for pregnancy married adolescents and not a counterfactual comparison to adolescent girls in non-MMA locations. Written informed consent was obtained on an electronic consent form and participants were provided with a paper copy with study contact information. Data collection occured from November 10, 2023, to November 4, 2024.

Global Public Health

Trained female mentors who were familiar with MMA were used to recruit potential study participants. Girls who expressed interest in the study were assessed for eligibility. Those who were eligible were asked to provide contact details for study enrollment purposes. Sample size determination was estimated using a two-sided Wald Z-test accounting for repeated measures and clustering within groups and assuming no change for the primary outcome(s) within the comparison group. To detect an effect size of 10% between baseline and the 3-month follow-up, a sample size of 545 girls per group was estimated under the following conditions: power of 80%, two-sided 95% confidence interval, design effect of 1.2,10% loss to follow-up, and 10% non-response rate. Sample allocation between girls who were already pregnant and those at-risk for pregnancy was assigned using a ratio of 3:1 to power the study to detect pregnancy-related outcomes (see Fig 1).

**Data collection**

The baseline and 3-month follow-up surveys were conducted with both girls who were at risk for pregnancy and pregnant girls who met eligibility criteria. The surveys focused on preconception knowledge, practices, and beliefs. The postpartum survey was administered to participants who were pregnant at baseline or became pregnant between baseline and the 3-month follow-up. The postpartum survey was conducted approximately 8-weeks after delivery and focused on ANC, PNC, behaviors during pregnancy, birth location, and PPFP uptake. Surveys were administered electronically by trained

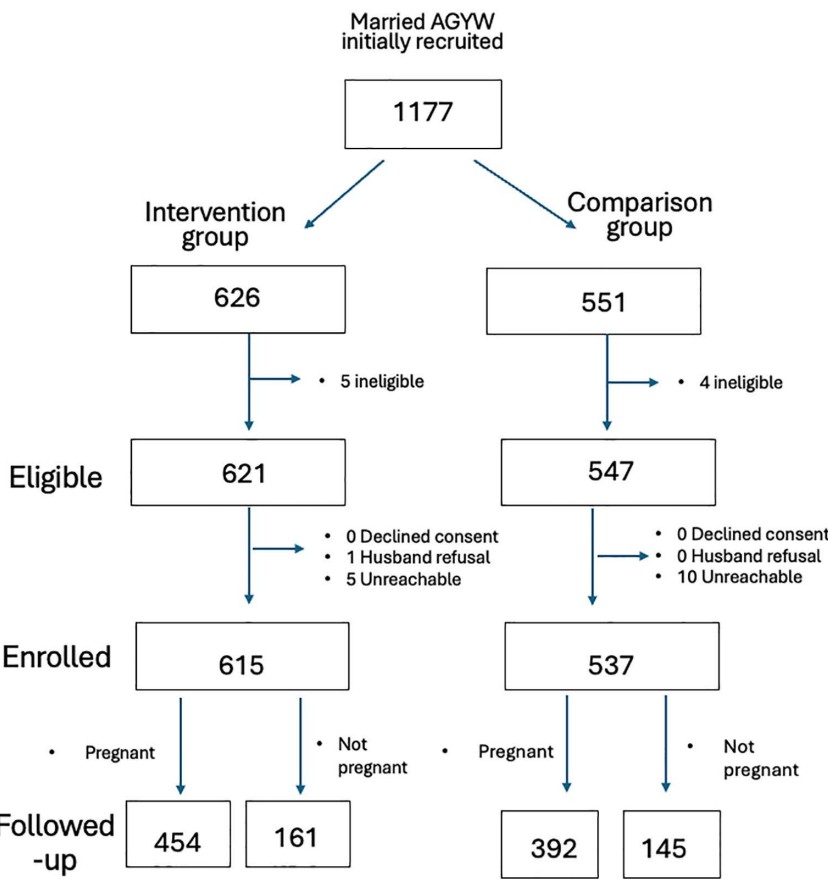

**Fig 1. Recruitment and sample selection.**

female enumerators and conducted in English and Hausa languages. Participants received $2 USD on completion of the 3-month and postpartum follow-up surveys.

## Data analysis

Several outcome measures were used to examine pregnancy and postpartum-related behaviors: *ANC attendance (attended or received any ANC during pregnancy from a skilled provider), # of ANC visits during pregnancy, gestational age in weeks at first ANC visit, consistent use of nutritional supplements (e.g., prenatal vitamin or folic acid) throughout pregnancy, place of birth, receiving PNC from a skilled provider within 48 hours of delivery,* and, *the participant or her husband doing something to prevent or space a subsequent pregnancy or birth after delivery.* Covariates included age, education, parity, and the multidimensional poverty index [25]. Statistical analysis of the postpartum data included univariate descriptive statistics (frequencies and means), bivariate tests of comparison between intervention and comparison groups (e.g., between group t-tests and chi-square tests of comparison), and multivariate linear and modified Poisson regression with robust standard errors [26,27]. In adjusted models, we also controlled for LGA to account for within group clustering. In addition, to increase comparability between groups, we weighted adjusted models by inverse probability weights (propensity scores). Variables that created 10% or greater difference between the unadjusted and adjusted effects were considered confounders and controlled for in the final model. Stata version 17 was used for all analysis.

## Ethics statement

Ethical approvals were obtained from Population Services International Research Ethics Board (#1735/2023), the Kaduna State Health Research Ethics Committee (# MOH/ADM/744/vol.1/1158) and Jigawa State Health Research Ethics Committee (# JGHREC/2023/176). Permissions were sought from the Local Government Health Secretariats at the LGA-level in both states. Participants were informed about the study objectives, procedures and the voluntary nature of participation before they provided signed written consent prior to participation. A waiver of parental consent for adolescent girls 15–17 years was granted by the ethical review committees since only married adolescent girls, who are considered emancipated, were involved. The study was registered as a clinical trial on 11-7-2024 at ClinicalTrials.gov (NCT06680765).

## Inclusivity in global research

Additional information regarding the ethical, cultural, and scientific considerations specific to inclusivity in global research is included in the Supporting Information (S1 File. *CONSORT Flowchart and* S1 Checklist. *Inclusivity in Global Research*).

## Results

Table 1 presents the demographic characteristics of the postpartum survey participants by study group. The total analytic sample was 846 adolescent girls (n = 454 intervention group, n = 392 comparison group). The average age across the sample was 18 (SD = 1.03). Three quarters (75.1%) of girls in the intervention group and 68.6% of girls in the comparison group were age 18 or 19. There was no significant difference in age by study group. Most girls were in monogamous marriages with only 16.5% of girls in the intervention group and 14.3% in the comparison group reporting polygamous relationships. There was no significant difference in relationship type by study group. Among the study sample, 52.5% were from Kaduna state and 47.5% from Jigawa state. Jigawa state had a slightly higher proportion of participants in the intervention group compared to the comparison group, 52.6% vs. 41.6%, and Kaduna state had a higher proportion of participants in the comparison group vs. the intervention group (58.4% vs. 47.4%, p < 0.001). Most study participants were of Islamic religion (92.9%). All participants in the intervention group reported Islam as their religion compared to 84.7% in the comparison group (p < 0.001). For education, 10.8% of girls in the intervention group were in-school at the time of the study compared to 18.4% in the comparison group (p = 0.002). Among girls not in-school, 57% in the intervention group

**Table 1. Demographic characteristics of study participants by study group.**

| | Intervention n (%)/ mean (SD) (n = 454) | Comparison n (%)/ mean (SD) (n = 392) | Total n (%)/ mean (SD) (N = 846) | Test of significance | p-value |
|---|---|---|---|---|---|
| **Age category** | | | | | |
| 15 years | 15 (3.3%) | 13 (3.3%) | 28 (3.3%) | Pearson chi2 = 6.41 | p = 0.170 |
| 16 years | 22 (4.9%) | 30 (7.6%) | 52 (6.2%) | | |
| 17 years | 76 (16.7%) | 80 (20.4%) | 156 (18.4%) | | |
| 18 years | 186 (41.0%) | 136 (34.7%) | 322 (38.1%) | | |
| 19 years | 155 (34.1%) | 133 (33.9%) | 288 (34.0%) | | |
| **Relationship type** | | | | | |
| Monogamous | 379 (83.5%) | 336 (85.7%) | 715 (84.5%) | Pearson chi2 = 0.80 | p = 0.370 |
| Polygamous | 75 (16.5%) | 56 (14.3%) | 131 (15.5%) | | |
| **State** | | | | | |
| Kaduna | 215 (47.4%) | 229 (58.4%) | 444 (52.5%) | Pearson chi2 = 10.32 | p < 0.001 |
| Jigawa | 239 (52.6%) | 163 (41.6%) | 402 (47.5%) | | |
| **Religion** | | | | | |
| Christianity | 0 (0.0%) | 60 (15.3%) | 60 (7.1%) | Pearson chi2 = 74.79 | p < 0.001 |
| Islam | 454 (100%) | 332 (84.7%) | 786 (92.9%) | | |
| **Education** | | | | | |
| Currently in school | 49 (10.8%) | 72 (18.4%) | 121 (14.3%) | Pearson chi2 = 9.84 | p = 0.002 |
| **Highest level of schooling (not currently in school)** | | | | | |
| None | 37 (9.1%) | 79 (24.7%) | 116 (16.0%) | Pearson chi2 = 37.16 | p < 0.001 |
| Primary | 92 (22.7%) | 44 (13.7%) | 136 (18.8%) | | |
| Secondary | 231 (57.0%) | 158 (49.4%) | 389 (54.7%) | | |
| Higher | 4 (0.9%) | 5 (1.6%) | 9 (1.2%) | | |
| Islamiyah school | 41 (10.1%) | 34 (10.6%) | 75 (10.3%) | | |
| **Multi-dimensional poverty index (% deprived households)** | 330 (73.0%) | 254 (64.8%) | 584 (69.0%) | Pearson chi2 = 6.13 | p = 0.013 |
| **Previous pregnancy** | 156 (34.4%) | 114 (29.1%) | 270 (31.9%) | Pearson chi2 = 2.69 | p = 0.110 |
| **Average age (years) at first pregnancy among those with previous pregnancy** | 15.98 (1.01) | 16.17 (1.66) | 16.06 (1.33) | Two-sample t test | p = 0.251 |

Notes. SD = standard deviation; less than 10% missing on frequencies/means.

and 49.4% in the comparison group had completed secondary education. The proportion of girls with no education was higher in the comparison group compared to the intervention group, 24.7% vs. 9.1%. Primary education also differed by group with more girls in the intervention group having completed primary education (22.7% vs. 13.7%). Differences in education levels achieved by study group were statistically significant (p < 0.001). Most participants (69%) were poor according to the multidimensional poverty index, with a slightly higher proportion of deprived households in the intervention group than the comparison group, 73% vs. 64.8% (p = 0.013). Nearly 30% of the sample had been pregnant prior to their current pregnancy. The average age at first pregnancy was 16.

Tables 2 and 3 present bivariate regression results for key pregnancy and postpartum indicators. The mean gestational age at the first ANC visit was 4.9 months for the intervention group and 5.3 months for the comparison group. Compared to the comparison group, girls in the intervention group sought ANC earlier by 0.53 months (95%CI: [-0.82, -0.25]). The

**Table 2. Bivariate analysis key pregnancy and postpartum variables by study group.**

|  | Mean (SD) | Beta | 95% CI | p-value |
|---|---|---|---|---|
| **Gestational age (months) at time of first ANC visit** |  |  |  |  |
| Comparison | 5.27 (5.05-5.50) | Ref | Ref | Ref |
| Intervention | 4.68 (4.54-4.82) | -0.53 | -0.82, -0.25 | p<0.001 |
| **# of ANC visits during pregnancy** |  |  |  |  |
| Comparison | 3.90 (3.66-4.14) | Ref | Ref | Ref |
| Intervention | 5.01 (4.84-5.18) | 1.11 | 0.81, 1.40 | p<0.001 |

Notes. SD=standard deviation; CI=confidence interval; Ref/ref=reference group; ANC=antenatal care.

**Table 3. Bivariate analysis for key pregnancy and postpartum variables by study group.**

|  | n (%) | IRR | 95% CI | p-value |
|---|---|---|---|---|
| **Attended ANC during pregnancy (No=ref)** |  |  |  |  |
| Comparison | 214 (54.6%) | Ref | Ref | Ref |
| Intervention | 432 (95.4%) | 1.75 | 1.59, 1.92 | p<0.001 |
| **Used nutritional supplements during the whole pregnancy (No=ref)** |  |  |  |  |
| Comparison | 236 (60.2%) | Ref | Ref | Ref |
| Intervention | 398 (88.8%) | 1.47 | 1.35, 1.61 | p<0.001 |
| **Place of birth (Home=ref)** |  |  |  |  |
| Comparison | 100 (28.99%) | Ref | Ref | Ref |
| Intervention | 318 (78.91%) | 2.72 | 2.29, 3.23 | p<0.001 |
| **Attended PNC (No=ref)** |  |  |  |  |
| Comparison | 151 (46.75%) | Ref | Ref | Ref |
| Intervention | 227 (58.81%) | 1.26 | 1.09, 1.45 | p<0.001 |
| **Girl or husband doing something to avoid pregnancy or delay pregnancy following delivery (No=ref)** |  |  |  |  |
| Comparison | 78 (19.95%) | Ref | Ref | Ref |
| Intervention | 215 (47.57%) | 2.38 | 1.91, 2.97 | p<0.001 |

Notes. IRR=incident rate ratio; CI=confidence interval; Ref/ref=reference group; ANC=antenatal care; PNC=postnatal care.

mean number of ANC visits during pregnancy was 1.11 times higher for girls in the intervention group vs. the comparison group (95%CI: [0.81, 1.40]). Both gestational age at first ANC visit and number of ANC visits were significantly associated with intervention exposure at p<0.001.

Nearly 95% of girls in the intervention group attended any ANC visit with a skilled provider compared to only 55% in the comparison group. The prevalence ratio for attending ANC was 1.75 (95%CI: [1.59, 1.92]) times greater for girls in the intervention group compared to the comparison group. Almost 90% of girls in the intervention group reported using a nutritional supplement throughout pregnancy compared to 60% of girls in the comparison group. The prevalence ratio for consistently using a nutritional supplement during pregnancy was 1.47 (95%CI: [1.35, 1.61]) times greater for girls exposed to the intervention compared to the comparison group. Almost 80% of girls exposed to the intervention reported facility-based birth compared to just under 30% in the comparison group. The prevalence ratio for a facility-based birth was 2.72 (95%CI: [2.29, 3.23]) times greater for girls exposed to the intervention compared to the comparison group. Slightly less than 60% of girls in the intervention group reported receiving PNC within 48 hours of delivery compared to 47% of girls in the comparison group. The prevalence ratio for receiving PNC was 1.26 (95%CI: [1.09, 1.45]) times greater among the intervention group compared to the comparison group. Nearly half (48%) of girls in the intervention group reported that they or their husband were doing something post-delivery to prevent or space future pregnancies compared

to just 20% of girls in the comparison group. The prevalence ratio for PPFP was 2.38 (95%CI: [1.91, 2.97]) times greater for girls exposed to the intervention compared to the comparison group. All outcomes were statistically significantly associated with intervention exposure at p<0.001.

Tables 4, 5, 6 present adjusted linear and modified Poisson regression models with robust standard errors, controlling for education, age, parity, and the multidimensional poverty index. LGA was also included in the models as a fixed effect to account for potential within group clustering. In adjusted models, the statistically significant associations detected in crude models remained statistically significant. Girls in the intervention group attended their first ANC visit at an earlier gestational age compared to girls in the comparison group (β: -0.54; 95%CI: [-0.89, -0.21]) and the number of ANC visits was higher by 1.39 (95%CI: [1.01, 1.77]).

The prevalence ratio for attending any ANC visit was 1.72 (95%CI: [1.56, 1.90]) times greater for girls exposed to the intervention compared to girls in the comparison group.

Compared to girls in the comparison group, the prevalence ratio was 1.49 (95%CI: [1.35, 1.64]) times greater for continuous nutritional supplementation during pregnancy for girls in the intervention group. Place of birth was also significantly associated with intervention exposure with a prevalence ratio of 2.44 (95%CI: [2.01, 2.96]) for facility-based birth. Girls in the intervention group had a substantially higher prevalence ratio for PPFP compared to girls in the comparison group (IRR: 3.17; 95% CI: [2.48, 4.05]) and attending PNC was 1.48 (95%CI: [1.25, 1.76]) times greater for girls exposed to the intervention vs. girls in the comparison group.

## Discussion

This study presents results from an adolescent-friendly MNCH intervention nested within a larger SRH-focused intervention targeting married pregnant adolescent girls aged 15–19 in northern Nigeria where adolescent pregnancy rates are high and use of perinatal and postnatal services low. The MNCH component was co-designed with the target population and aimed to increase knowledge, attitudes, and behaviors related to preconception care and healthy behaviors during

**Table 4. Adjusted regression analysis for pregnancy variables by study group.**

| | Gestational age at first ANC visit | | | Attended ANC | | | # ANC visits | | |
|---|---|---|---|---|---|---|---|---|---|
| | Beta | 95% CI | p-value | IRR | 95% CI | p-value | Beta | 95% CI | p-value |
| Intervention group (comparison=ref) | **-0.54** | **-0.89 -0.21** | **p<0.001** | 1.72 | 1.56, 1.90 | **p<0.001** | 1.39 | 1.01, 1.77 | **p<0.001** |
| Previous birth (no=ref) | 0.11 | -0.17, 0.40 | p=0.451 | **0.91** | **0.85, 0.99** | **p=0.023** | -0.22 | -0.56, 0.11 | p=0.189 |
| Multi-dimensional Poverty Index (no=ref) | | | | | | | | | |
| Yes | 0.13 | -0.13, 0.39 | p=0.451 | **0.90** | **0.83, 0.97** | **p=0.007** | -0.10 | -0.41, 0.20 | p=0.519 |
| Education (none=ref) | | | | | | | | | |
| Primary | -0.03 | -0.56, 0.50 | p=0.913 | **0.85** | **0.75, 0.97** | **p=0.015** | 0.03 | -0.54, 0.61 | p=0.902 |
| Secondary | -0.29 | -0.80, -0.22 | p=0.265 | **0.80** | **0.71, 0.91** | **p<0.001** | 0.27 | -0.29 0.83 | p=0.348 |
| Tertiary | **-1.02** | **-1.93, -0.11** | **p=0.029** | 0.76 | 0.57, 1.02 | p=0.064 | **1.63** | **0.45, 2.80** | **p=0.007** |
| Islamic school | -0.16 | -0.84, 0.53 | p=0.650 | **0.73** | **0.62, 0.87** | **p<0.001** | 0.40 | -0.30, 1.12 | p=0.260 |
| Age (15 years=ref) | | | | | | | | | |
| 16 years | 0.45 | -0.26, 1.15 | p=0.218 | 1.03 | 0.81, 1.30 | p=0.818 | 0.09 | -0.81, 1.00 | p=0.850 |
| 17 years | 0.00 | -0.61, 0.61 | p=0.995 | 1.02 | 0.84, 1.25 | p=0.836 | -0.11 | -0.83, 0.63 | p=0.774 |
| 18 years | 0.36 | -0.22, 0.93 | p=0.221 | 1.08 | 0.89, 1.31 | p=0.419 | 0.40 | -0.65, 0.73 | p=0.909 |
| 19 years | 0.07 | -0.50, 0.65 | p=0.803 | 1.13 | 0.93, 1.36 | p=0.220 | 0.46 | -0.25, 1.18 | p=0.204 |
| LGA | 0.26 | -0.22, 0.73 | p=0.291 | **1.32** | **1.19, 1.46** | **p<0.001** | -0.15 | -0.69, 0.39 | P=0.593 |

Notes. IRR = incidence rate ratio; CI = confidence interval; ref = reference; ANC = antenatal care; LGA = local government area.

[1]All models weighted by propensity score (inverse probability weighting).

PLOS Global Public Health

**Table 5. Adjusted regression analysis for pregnancy and postpartum variables by study group.**

| | Nutritional supplements during pregnancy | | | Place of birth | | | PPFP | | |
|---|---|---|---|---|---|---|---|---|---|
| | IRR | 95% CI | p-value | IRR | 95% CI | p-value | IRR | 95% CI | p-value |
| Intervention group (comparison=ref) | **1.49** | **1.35, 1.64** | **p<0.001** | **2.44** | **2.01, 2.96** | **p<0.001** | **3.17** | **2.48, 4.05** | **p<0.001** |
| Previous birth (no=ref) | 0.96 | 0.89, 1.04 | p=0.319 | 0.97 | 0.85, 1.11 | p=0.666 | 0.92 | 0.75, 1.12 | p=0.394 |
| Multi-dimensional Poverty Index (no=ref) | | | | | | | | | |
| Yes | 0.94 | 0.86, 1.02 | p=0.152 | 0.97 | 0.85, 1.10 | p=0.685 | 1.10 | 0.89, 1.37 | p=0.360 |
| Education (none=ref) | | | | | | | | | |
| Primary | **0.80** | **0.70, 0.92** | **p<0.001** | 0.88 | 0.70, 1.10 | p=0.269 | 0.76 | 0.53, 1.10 | p=0.148 |
| Secondary | **0.78** | **0.69, 0.89** | **p<0.001** | 0.94 | 0.78, 1.16 | P=0.570 | 0.82 | 0.56, 1.18 | p=0.284 |
| Tertiary | 0.81 | 0.62, 1.06 | p=0.126 | 1.30 | 0.91, 1.87 | p=0.147 | 1.43 | 0.82, 2.52 | p=0.207 |
| Islamic school | **0.74** | **0.62, 0.88** | **p<0.001** | 0.80 | 0.63, 1.03 | p=0.090 | **0.57** | **0.34, 0.95** | **p=0.032** |
| Age (15 years=ref) | | | | | | | | | |
| 16 years | 0.93 | 0.71, 1.22 | p=0.585 | 1.30 | 0.87, 1.92 | p=0.194 | 0.88 | 0.52, 1.48 | p=0.623 |
| 17 years | 0.98 | 0.80, 1.21 | p=0.891 | 1.24 | 0.86, 1.78 | p=0.252 | 0.88 | 0.56, 1.39 | p=0.587 |
| 18 years | 1.05 | 0.87, 1.28 | p=0.588 | 1.20 | 0.84, 1.72 | p=0.309 | 0.75 | 0.48, 1.15 | p=0.188 |
| 19 years | 1.11 | 0.91, 1.35 | p=0.288 | 1.20 | 0.84, 1.73 | p=0.310 | 0.67 | 0.44, 1.04 | p=0.073 |
| LGA | **1.23** | **1.11, 1,37** | **p<0.001** | **1.51** | **1.25, 1.82** | **p<0.001** | 1.33 | 0.92, 1.91 | p=0.124 |

Notes. IRR=incidence rate ratio; CI=confidence interval; ref=reference; PPFP=postpartum family planning; LGA=local government area.

[1]All models weighted by propensity score (inverse probability weighting).

**Table 6. Adjusted regression analysis for pregnancy and postpartum variables by study group (continued).**

| | Attended PNC | | |
|---|---|---|---|
| | IRR | 95% CI | p-value |
| Intervention group (comparison=ref) | **1.48** | **1.25, 1.76** | **p<0.001** |
| Previous birth (no=ref) | 0.88 | 0.74, 1.03 | p=0.120 |
| Multi-dimensional Poverty Index (no=ref) | | | |
| Yes | 0.95 | 0.82, 1.11 | p=0.534 |
| Education (none=ref) | | | |
| Primary | 0.93 | 0.66, 1.19 | p=0.635 |
| Secondary | 1.03 | 0.78, 1.36 | p=0.800 |
| Tertiary | 1.13 | 0.68, 1.87 | p=0.627 |
| Islamic school | 1.35 | 0.95, 1.93 | p=0.093 |
| Age (15 years=ref) | | | |
| 16 years | 1.20 | 0.71, 2.04 | p=0.493 |
| 17 years | 1.08 | 0.66, 1.76 | p=0.755 |
| 18 years | 1.24 | 0.78, 1.98 | p=0.354 |
| 19 years | 1.19 | 0.75, 1.92 | p=0.456 |
| LGA | 0.81 | 0.63, 1.05 | p=0.118 |

Notes. IRR=incidence rate ratio; CI=confidence interval; ref=reference; PPFP=postpartum family planning; LGA=local government area.

[1]All models weighted by propensity score (inverse probability weighting).

pregnancy and postpartum. It included educational sessions, co-facilitated by trained female community mentors and government ANC providers at local facilities that offered girls an opportunity to learn about the importance of ANC and PNC, the benefits of facility-based birth and PPFP and to build trust with service providers. This was complimented by direct outreach to husbands about the value of ANC, follow-up support from the female community mentors during pregnancy and training providers on how to engage adolescents, build trust and provide adolescent-friendly care. Results from the postpartum survey suggest that adolescent girls exposed to the MNCH component were statistically significantly more likely to attend their first ANC visit at an earlier gestational age and attend more ANC visits compared to girls in the comparison group who did not receive the MNCH component, net of the effects of age, education, parity, and poverty. In addition, girls exposed to the MNCH component consistently used nutritional supplements throughout pregnancy at higher rates than girls in the comparison group and were more likely to give birth in a facility, receive PNC, and use PPFP.

These results provide promising evidence about the ability to reach a high-need population of married adolescents by integrating an MNCH component within a larger SRH-focused intervention. As indicated in the systematic review by Sabet et al., there is a significant evidence gap on interventions that focus on increasing uptake and use of pregnancy-related health services among pregnant adolescents, making it challenging to compare our results to similar interventions among the same population [13]. Nonetheless, our intervention approach adds to available evidence on how to create effective programs to improve maternal and reproductive health behaviors and outcomes for married adolescents and WRA more generally, and provides evidence on the effectiveness of using a co-design process to create an intervention that addresses the unique needs of the target population. Mhango et al. assessed the effectiveness of an adolescent co-designed intervention that focused on mental health for pregnant adolescent girls in Malawi [28]. The intervention consisted of psychosocial educational sessions with CHWs, and results were overwhelmingly positive with significant improvements reported across all mental health outcomes. Both our MNCH intervention components and the intervention assessed by Mhango et al., are examples of programs that reinforced principles of respectful maternal care by responding to the voices of pregnant girls and recognizing them as valuable collaborators in program design, which likely contributed to the success of the interventions.

Our approach also makes use of other evidence-based strategies, including group-based approaches, engaging with husbands, and trust building with providers. According to Sabet et al., group-based approaches may be effective platforms to engage adolescents as they provide opportunities to share experiences and build a peer-group [13]. The educational sessions, although limited in number, provided adolescent girls in similar stages of pregnancy a chance to learn together and be part of a group in which all members were concerned with having a healthy pregnancy. Although evidence on engaging husbands of adolescents in maternity care is scarce, a systematic review on pregnancy-related intervention by Tokhi et al. demonstrates that engaging husbands in the pregnancy care process is associated with improved ANC attendance, increased likelihood of facility-based birth, and seeking PNC [29]. Our study supports these findings specifically within the sub-group of pregnant adolescent.

It is important to acknowledge that the success of the MNCH component was likely due to the multilevel design that spanned the continuum of care. Drawing on lessons learned from the MMA program and other evidence-based programming, the MNCH component was designed to address barriers at different levels of the socio-ecological model. Not only did the intervention target individual care-seeking behaviors of pregnant girls, but also interpersonal determinants through fostering supportive relationships and communication between girls and their husbands, as well as girls and the female community mentors. At the facility-level, provider capacity to provide adolescent-friendly care and establishing trust between providers and patients were strengthened. Our approach supports the evidence that multilevel interventions are more likely to lead to more substantial changes in behaviors than single-level interventions [30].

Our findings add to the growing body of evidence demonstrating the value of integrated approaches. While the comparison group received education and access to family planning services, the intervention group received more comprehensive support across the maternal and reproductive health spectrum, which contributed to significantly higher PPFP

uptake. This suggests a holistic model, which provides both immediate support (e.g., promoting healthy behaviors during pregnancy and ensuring access to quality ANC) and anticipatory guidance (e.g., counseling on desired family size and access to modern contraception), better aligns with girls' evolving needs and preferences. Continuum of care models are well-suited for health system structures where complimentary services are provided by the same providers or within the same facility.

While this intervention showed significant improvements for the intervention group, more work is needed to scale effective approaches to reach all girls in need. Additionally, significant gaps still exist in understanding what approaches are most effective for behavior change, cost-effective, and sustainable for this population. In our analysis, LGA was a significant predictor of ANC and PNC attendance. This may be influenced by distance from facilities, or other unique elements of local communities that the analysis was not able to isolate such as localized social norms, facility quality, or influential leaders. More research is needed to understand how community-specific factors affect service utilization [31]. Adolescent pregnant girls in Nigeria continue to face barriers to accessing and receiving high quality maternal care. The most prominent barriers include access-related barriers, social stigma, lack of decision-making power, provider bias, and health system constraints. Pregnant adolescent girls especially in the conservative religious and cultural context of northern Nigeria remain a challenging group to identify, reach, and engage. While the MNCH intervention component was effective at improving knowledge and healthcare seeking behaviors for pregnant adolescent girls and tried to address more interpersonal barriers including provider bias and gender power-imbalances, barriers at the structural and policy levels take longer to change and must be addressed to have a sustained impact on adolescent girls' health [32].

This study is not without limitations. The study is subject to response bias and recall bias due to self-reporting and interviewer-administered tools. Sample size calculations were made based on the entire sample, not for the sub-sample of pregnant women. It is possible that the sample size for the sub-analysis was not powered enough to detect effects. We were unable to capture the existence of other SRH interventions that may have occurred at the same time as the intervention presented here which could confound the study results. The study was conducted in LGAs where the parent intervention, MMA, had been implemented, limiting our ability to compare results to the true counterfactual among girls in non-MMA areas. Because the MNCH component focused on addressing knowledge, attitudes and behaviors, it cannot be directly linked to improved health outcomes. Although the intervention had multiple activities, we are unable to identify the effectiveness of each activity, which is also a limitation of the study design.

## Conclusion

Complications from pregnancy during adolescence are a leading cause of death for adolescent girls in Nigeria. Using a co-design approach, we developed an adolescent-friendly MNCH intervention component to improve knowledge and pregnancy care-seeking behaviors for married pregnant adolescent girls and girls at risk for pregnancy in northern Nigeria. The component was associated with earlier ANC initiation, a greater number of ANC visits attended, use of nutritional supplements during pregnancy, facility-based birth, receipt of PNC, and PPFP uptake. The positive results demonstrate the potential impact that co-designed adolescent-friendly interventions can have on pregnancy-related health behaviors for a vulnerable and underserved population.

## Supporting information

**S1 File. CONSORT Flowchart (Hopewell S, Chan A-W, Collins GS, Hróbjartsson A, Moher D, Schulz KF, et al. (2025) CONSORT 2025 statement: Updated guideline for reporting randomised trials.** PLoS Med 22(4): e1004587). (DOCX)

**S2 File. Human Centered Design Process.** (DOCX)

**S1 Checklist. Inclusivity in Global Research.**
(DOCX)

## Acknowledgments

We would like to acknowledge the study participants, Viable Knowledge Masters, SFH Nigeria teams who implemented the project, government partners in Kaduna and Jigawa states, and the girls who consented to participate in the study.

## Author contributions

**Conceptualization:** Erica Felker-Kantor, Mary Phillips, Meghan Cutherell, Abednego Musau.

**Data curation:** Stacy Lois, Alhaji Bulama.

**Formal analysis:** Erica Felker-Kantor.

**Funding acquisition:** Meghan Cutherell, Roselyn Odeh.

**Methodology:** Erica Felker-Kantor, Mary Phillips, Meghan Cutherell, Abednego Musau.

**Project administration:** Mary Phillips, Meghan Cutherell, Stacy Lois, Abednego Musau.

**Resources:** Meghan Cutherell, Roselyn Odeh.

**Supervision:** Roselyn Odeh, Stacy Lois, Alhaji Bulama, Abednego Musau.

**Validation:** Alhaji Bulama.

**Writing – original draft:** Erica Felker-Kantor.

**Writing – review & editing:** Erica Felker-Kantor, Mary Phillips, Meghan Cutherell, Roselyn Odeh, Stacy Lois, Alhaji Bulama, Abednego Musau.

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
