## [Decision Letter · Decision Letter 0]

24 Sep 2025

PGPH-D-25-01784

Results from a maternal, newborn and child health program targeting pregnant, married adolescent girls in northern Nigeria

Dear Dr. Felker-Kantor,

Thank you for submitting your manuscript to PLOS Global Public Health. After careful consideration, we feel that it has merit but does not fully meet PLOS Global Public Health’s publication criteria as it currently stands. Therefore, we invite you to submit a revised version of the manuscript that addresses the points raised during the review process.

We look forward to receiving your revised manuscript.

Kind regards,

Dickson Abanimi Amugsi, PhD

Academic Editor

Journal Requirements:

i. State the initials, alongside each funding source, of each author to receive each grant.

ii. State what role the funders took in the study. If the funders had no role in your study, please state: “The funders had no role in study design, data collection and analysis, decision to publish, or preparation of the manuscript.”

3. Please ensure that your Ethics Statement is available in its entirety at the beginning of your Methods section, under a subheading 'Ethics Statement'. It must include:

1) The name(s) of the Institutional Review Board(s) or Ethics Committee(s)

2) The approval number(s), or a statement that approval was granted by the named board(s)

3) (for human participants/donors) - A statement that formal consent was obtained (must state whether verbal/written) OR the reason consent was not obtained (e.g. anonymity).

NOTE: If child participants, the statement must declare that formal consent was obtained from the parent/guardian.

4. Please provide separate figure files in .tif or .eps format.

5. We have noticed that you have uploaded Supporting Information files, but you have not included a list of legends. Please add a full list of legends for your Supporting Information files after the references list.

6. Please note that your Data Availability Statement is currently missing the repository name and/or the DOI/accession number of each dataset OR a direct link to access each database. If your manuscript is accepted for publication, you will be asked to provide these details on a very short timeline. We therefore suggest that you provide this information now, though we will not hold up the peer review process if you are unable.

Additional Editor Comments (if provided):

Reviewer #1:

Reviewer #2:

Reviewers' comments:

Reviewer's Responses to Questions

**Comments to the Author**

1. Does this manuscript meet PLOS Global Public Health’s publication criteria?

Reviewer #1: Partly

Reviewer #2: Partly

2. Has the statistical analysis been performed appropriately and rigorously?

Reviewer #1: No

Reviewer #2: No

3. Have the authors made all data underlying the findings in their manuscript fully available (please refer to the Data Availability Statement at the start of the manuscript PDF file)?

Reviewer #1: Yes

Reviewer #2: Yes

4. Is the manuscript presented in an intelligible fashion and written in standard English?

Reviewer #1: Yes

Reviewer #2: Yes

Reviewer #1: This paper summarizes results from a longitudinal quasi-experimental study evaluating the effectiveness of a maternal, newborn and child health (MNCH) intervention on pregnancy and postpartum outcomes among married adolescent girls in northern Nigeria. The paper is well written from the statistical perspective. However, there are some clarifications needed.

Although comparative , the study is non randomized using some LGA sites as the intervention sites and others as the control or comparison. The study appears large enough.

1. Since treatment had been pre assigned did the investigators consider any matching such as propensity score matching or other method to assure further comparability of the samples to each other? With a 3:1 assignment this may be challenging, but can be done.

2. The sample size calculation was reasonable. However the dropout or loss rate for each group is higher than 10%. Were any retention or sample augmentation strategies implemented into the design? This should be added as a limitation in the discussion section.

The data analysis is routine in this setting using between group t-tests and chi-square tests of comparison), and multivariate linear and modified Poisson regression.

3. In the regression please explain the controlling for the LGA. The quantitation structure for this variable is not clear.

Reviewer #2: Thank you for submitting your manuscript that describes an intervention aimed at addressing signficant barriers to manternal and child health care for adolescents, especially in Northern Nigeria.

BACKGROUND:

The authors provide sufficient justfication for conducting this study in Northern Nigeria, and make a good case for the manuscripts contribution to the literature on SRH care for adolescents in similar setttings. Kudos to the team.

Line 164: The authors highlight the importance of recognizing adolescents as partners in designing health programs. Given that this is an important component of the intervention, I suggest the authors include more information about the HCD process, as a supplement.

METHODS:

Line 250: I suggest the authors provide more information on the criteria/method used in selecting the intervention and comparison sites within the study areas

Line 257/258: The authors indicate their interest in measuring incremental effectiveness. I suggest the authors include some information on the outcomes of the MMA component alone, to provide some bases/context for understanding the intended addtional effect of the MNCH component.

Line 259: It is unclear who the target population is for the MNCH component of the intervention. Do the authors mean that adolescent girls who were already recruited for the MMA component were then recruited again for the MNCH component when they got pregnant at the 3-month follow-up, or did the authors provide both components of the intervention to a cohort of girls who met the eligibility criteria in the study areas?

Line 277: The sample size was estimated to detect an effect size between baseline and the 3-month follow-up. However, the authors indicate in Line 230 that this paper presents results from the post-partum survey. It is unclear how if the sample size calculated is indeed powered enough to detect effects post-partum. Some food for thought and clarity by the authors.

Line 285: Since the postpartum survey was conducted only for participants who were pregnant at baseline or at the 3-month follow-up, it will be important to present the baseline characteristics of these participants (e.g gestation of pregnancy). If possible, the authors could stratify their results by participants pregnant at baseline and those pregnant at the 3-month follow-up. I suspect metrics such as number of ANC visits and prevalence of ANC/PNC attendance might differ for these two groups, which provides an important indicator of the timing of the proposed intervention.

RESULTS

General comment: I suggest the authors provide information on the fidelity/dose/reach of the all aspects of the MNCH component in both the intervention and comparison groups. At present, this is assumed to be the same.

Line 327: Given that education was identified as an key determinant of access to care, and the relatively large diffence in education between the intervention and comparison groups, it might be useful to consider a sub-group analysis by educational status (education vs no education)

Line 338: If the mean gestational age at first ANC was 4.9 months, does this mean that at baseline (and at 3-months follow-up), all participants had not attended ANC? I think it is important to present a table of demographics for all participants (intervention and comparison groups) at baseline to provide more analytical context for the study.

Table 3: Given that LGA remains significant in ANC/PNC attendance (prevalence and number of ANC visits), and the fact that distance to facility of care was highlighed as one of the barriers to accessibility, I suggest the authors add this to their discussion and suggest ways that further studies might account for this salient factor.

**Do you want your identity to be public for this peer review?** For information about this choice, including consent withdrawal, please see our Privacy Policy

Reviewer #1: No

Reviewer #2: No

*Cannot connect to Ginger* Check your internet connection

600/12095 free characters checked.?>Go Premium

---

## [Decision Letter · Decision Letter 1]

21 Dec 2025

Results from a maternal, newborn and child health program targeting pregnant, married adolescent girls in northern Nigeria

PGPH-D-25-01784R1

Dear Dr. Felker-Kantor,

We are pleased to inform you that your manuscript 'Results from a maternal, newborn and child health program targeting pregnant, married adolescent girls in northern Nigeria' has been provisionally accepted for publication in PLOS Global Public Health.

Best regards,

Julia Robinson

Executive Editor

Reviewer Comments (if any, and for reference):

Reviewer's Responses to Questions

**Comments to the Author**

Reviewer #1: All comments have been addressed

Reviewer #2: All comments have been addressed

publication criteria?

Reviewer #1: (No Response)

Reviewer #2: Yes

3. Has the statistical analysis been performed appropriately and rigorously?

Reviewer #1: (No Response)

Reviewer #2: Yes

4. Have the authors made all data underlying the findings in their manuscript fully available (please refer to the Data Availability Statement at the start of the manuscript PDF file)?

Reviewer #1: (No Response)

Reviewer #2: Yes

5. Is the manuscript presented in an intelligible fashion and written in standard English?

Reviewer #1: (No Response)

Reviewer #2: Yes

Reviewer #1: (No Response)

Reviewer #2: (No Response)

**Do you want your identity to be public for this peer review?** For information about this choice, including consent withdrawal, please see our Privacy Policy

Reviewer #1: No

Reviewer #2: No
